# Understanding the Effects of Virtual Reality System Usage on Spatial Perception: The Potential Impacts of Immersive Virtual Reality on Spatial Design Decisions

Sahand Azarby * and Arthur Rice

College of Design, North Carolina State University, Raleigh, NC 27607, USA
* Correspondence: sazarby@ncsu.edu

**Abstract:** The main component of any Virtual Reality (VR) system is the human user. The ways in which a VR system shapes human experience can affect design outcomes. This research explores the differences in spatial perception between an immersive Virtual Reality Interactive Environment (IVRIE) and traditional Virtual Reality (also known as a desktop-based Virtual Reality system, abbreviated herein as the DT system). Spatial perception and the cognition of the spatial factors of virtual spaces were studied based on different features of the two systems, including the sense of immersion, forms of interaction, experience of human scale, and movement through virtual spaces. This study focused on determining how users' spatial decision making and performance were affected by differences in spatial perception created by the IVRIE and DT systems. Factors examined included the differences between and within the two virtual systems, based on differences in texture, system usage sequence, and the complexity of the experiential/spatial guidelines. Descriptive and inferential statistical testing using quantitative and qualitative data were used to find differences in spatial perception and decision making. The results showed significant space size variations produced by participants between and within the two different VR systems.

**Keywords:** virtual reality; immersive virtual environment; spatial perception; spatial decision making; immersion; interaction; user experience; spatial design

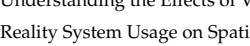



## 1. Introduction

Digital technology has drastically changed visualization and modeling techniques. Today, three-dimensional (3D) visualization methods and dynamic models have replaced analog visualization and static representations. The utilization of virtual environments for visualization and modeling has changed digital architectural representations from abstract and static to highly realistic, potentially providing an immersive experience on multiple spatial and temporal scales [1].

This study was motivated by the long-term goals of exploring and identifying the potential of VR as a digital design environment/tool for architectural design, including its use in design learning, teaching, and practice. The overall objective was to better understand differences in users' perception of the spatial factors of design in an immersive Virtual Reality Interactive Environment (IVRIE), as compared to the spatial perception obtained using a desktop-based semi-immersive virtual environment system (also known as traditional VR or a desktop-based Virtual Reality system (DT system)).

VR as a visualization tool has experienced a recent boom in the professional and educational design fields [2]. In many studies, VR, its characteristics, and possible capabilities have been highlighted in connection with digital design imperatives. VR as a visualization tool or design environment can be integrated into the design process from the initial design phases to the most advanced stages, affecting conventional design methods [3].

Improvements in information representation in VR can lead to more qualitative and perceptually accurate representations of designs. In other words, VR enables the representation of spaces from a user's perspective by presenting 3D spatial information at full scale and creating the illusion of depth and immersion [4]. VR has the potential to enhance visual and design thinking and play a functional role in problem solving, unearthing design solutions and enabling users to produce more thoughtful and rational designs. Thus, the ability to visualize designs more accurately will offer designers the confidence to alter their approach to design when needed [5,6]. The hope is that as a representation medium, VR will enable a viewer to understand a proposed design solution and allow for a more meaningful critique [4,5].

VR has often been studied as a learning environment. Researchers have frequently concluded that it enhances the educational context of learning, improves the process of learning initial design concepts, and facilitates an understanding of spatial relationships [6,7]. In addition, studies have suggested that new frameworks for design pedagogy that integrate digital concepts as a unique body of knowledge, consisting of the relationship between digital architectural information and digital design skill, can influence the development of theoretical, computational, and cognitive approaches in design education and practice [5,8].

Based on the ability of immersive VR to provide a level of engagement that improves spatial awareness, the possibility of utilizing VR visualization capabilities as a design tool provides abundant research opportunities [6]. VR compares favorably to other media in terms of how it conveys spatial data to users. It presents spatial information more accurately and in greater quantity than conventional media, possibly improving viewers' spatial awareness and leading them to better understand their designs' 3D nature in a more efficient fashion [3,4,9,10].

Although research on immersive virtual environments has increased in the last decade, few represent comparative, quantitative, and user-centered studies exploring this new technology's impact on the design process and outcomes and the learning of design concepts [7,11]. Research focusing on the effectiveness of evaluations of different VR systems (mixed methods) and studies comparing the functionality of virtual environments from different provider systems are even scarcer. The efficiency of any virtual environment can only be established by comparing human perception within and between different virtual environments or between virtual and real environments [11–13]. Research based on human factors and how these environments transfer spatial data to users and addressing user experience and performance within such environments has underscored the effectiveness of virtual systems in maximizing perception of the spatial factors they include [11,13,14].

Thus, while VR systems can provide different levels of immersion and a variety of types of interaction with virtual objects, all likely to represent important benefits to designers seeking to understand the spatial factors of their designs within a virtual environment, an important question remains. Are there any fundamental differences and influences between fully immersive interactive and semi-immersive VR in terms of transferring spatial data to designers for spatial decision making?

Only by understanding which, to what extent, and in what circumstances VR systems are able to improve the spatial cognition and perception of users, leading them to more accurate and logical spatial decision making and performance, can the actual efficiency of the technological, environmental, and representational parameters of these systems that benefit design learning and practice be identified.

*Purpose Statement and Research Questions*

This study determined how users' spatial decision making and performance were affected by differences in spatial perception and the use of either a conventional desktop computer system running the SketchUp software package or an immersive IVRIE system running the VR Sketch program and utilizing a VR headset and hand controllers. The research questions were as follows:

- Does design in IVRIE using immersive presence and direct interaction with design elements impact a designer's understanding of scale/volume? If so, how does this change their understanding of and impact the design outcomes?
- If there is a change, to what degree does the user's perception of scale, size, depth, and distance differ between these two systems?
- If the degree of difference in user perception and performance is significant, do users' particular usage of virtual environments and the characteristics of virtual spaces play any role in those differences?
- If positive, do users have an awareness of virtual environments' different features in terms of perception and performance?

The hypotheses for this research study were as follows. First, the spatial character of the design, spatial structure, scale, and extent of the designed/created space will be different when using the IVRIE and DT systems. Second, the combination of a sense of full immersion and direct interaction with virtual design objects in IVRIE will facilitate a more intuitive conception of the spatial factors of design, fulfilling specific spatial criteria not offered by DT systems and providing a sense of semi-immersion and indirect interaction with design objects.

## 2. Research Background

### 2.1. Virtual Reality Environments

A VR-related literature review revealed four key features of virtual environments: presence, spatial perception, immersion, and interaction. The studies reviewed for this research tended to conclude that the integration of immersion with interaction is what constructs presence in a VR environment, leading and in other ways affecting spatial perception [11,15–18].

Presence: Presence, as one of the main features of the virtual world, has a critical role in synthesizing immersion and interaction. In virtual worlds, whenever the level of presence is high, the sensation of immersion is so strong that the interface seems to disappear and users lose all notion of interacting with a machine [9]. The strength of presence experienced in a virtual environment varies as a function of both individual differences in the user and characteristics of the virtual environment; thus, individual differences, traits, and abilities could enhance or detract from the presence experienced in a given virtual environment [19]. Conditions necessary for presence in a virtual environment include attention, involvement, and immersion. The definition of attention (specifically, selective attention) is the tendency to focus on selected information that is meaningful and of particular interest to the user [18,19]. Presence is the feeling of being present in a given environment, upon its perception. To perceive oneself inside an environment is the origin of a sense of presence, which is a complex awareness phenomenon [11,20]. Visual-based awareness can improve different categories of visual perception, such as organizational principles, proximity relationships, similarities, shape properties, and the figure–ground relationship [21]. Being present within the designed environment is related to various factors such as being able to move through the design and creating the design through physical gestures [10].

Spatial perception: Spatial perception in virtual worlds is based on the characteristics and simulation power of the virtual environment to create spatial presence. Additionally, along with the specifications of the virtual environment, in most studies, spatial perception was found to be related to user characteristics such as spatial ability, thinking, and cognition. Spatial perception may have a direct connection with users' spatial and imagery abilities and the way that spatial situation models are formed for them. The overall assumption is that individuals who can effectively process spatial arrangements find it easier to create a "mental model" of the spatial environment; thus, spatial ability may have a more critical role in spatial perception when using desktop systems because users need to expend more cognitive effort to interpret spatial relationships [22]. Since VR can assist designers by

providing a highly visual mechanism and creating an inherently spatial environment, its ability to improve spatial awareness and perception may be a clear benefit [2,23,24].

The level of detail within a VR simulation may impact users' accuracy in constructing mental images of a space and have different effects on the development of spatial perception [5]. The accuracy of distance perception and judgment may differ with the quality of the graphics (i.e., rendering effects) of the virtual simulation, as well as textures and lighting conditions [11,19].

Immersion: Immersion is a psychological state characterized by the perception of a person being enveloped by, included in, and interacting with an environment [25]. VR as a system or environment has been divided into two categories: semi-immersive and immersive. Although the origination of VR can be traced to the early 1960s, in the early 1990s, full-immersion VR environments were developed at the University of Illinois and University of Chicago for astrophysics applications, introducing novel tools for reaching new levels of cognition for complex and massive datasets [26].

In most studies, comparisons of the kind and amount of immersion and level of presence between fully and semi-immersive environments have been related to the types of interaction with virtual objects that is possible within these environments. In a semi-immersive VR environment, the user is partially immersed in the virtual world and cannot directly interact with existing objects. In contrast, fully immersed observers perceive that they are interacting directly (and not indirectly or remotely) with the environment; they feel as if they are a part of that environment. Semi-immersive environments can also be called monitor-based virtual environments or, for the purposes of the present research, desktop-based systems (DT systems) [19,27,28]. A fully immersive environment is more powerful at involving users and increasing the sense of presence. Although semi-immersive VR has proven to be an influential visualization tool, the sense of presence is significantly less than with immersive VR [6,29,30]. Categorizing immersion based on the sense mode demonstrates that fully immersive environments are more potent than those that are semi-immersive.

Immersion can be defined based on interaction, and in many studies, these two characteristics have been inseparable. Any interaction with the environment in a natural manner should increase immersion, and thus presence. Being truly immersed in an environment that responds to the user's actions and interactions plays a crucial role in transferring data from a design tool to the user [19,29].

Interaction: In most studies, interaction is defined in conjunction with immersion and highlighted as a factor that improves 3D understanding within virtual environments. Interaction is one of VR's advantages, enabling users to manipulate the virtual environment and have an active experience. The interaction and reflection between the designer and environment empower the designer to have control over the digital processes and become more informed about their design [31,32]. Since in interactive 3D models design objects can be described along with other existing objects, the size and scale of the project and other design elements is often more understandable. Due to the guidance these characteristics provide, users can improve their 3D understanding of the project and design [26].

Interaction with design representations is a fundamental factor in design and can be categorized as either external or internal. External interactions are traditional types of direct interactions with shapes and forms, also known as man–machine interactions. The ultimate goal of external interaction, along with feeling immersed in the VR environment, is to reflect time and space for users as if they are experiencing them through the process of imagination and information acquisition. Internal interactions are related to interactions with digital forms through the medium of specific digital environments, computational processes, or mechanisms [31,33]. In VR, interactions with virtual design objects and elements within the virtual environment are internal interactions. The term "interaction" has also been defined as a combination of navigation and manipulation. The interactivity of a virtual environment is its ability to enable users to change their viewpoint and simultaneously have the experience of navigation (e.g., wayfinding and orientation) and manipulation.

Here, manipulation means scene manipulation, in which users can change objects' relative positions, including full object and object position changes. Full object change includes all actions related to creating or deleting design objects, while object position change gives the user the ability to reposition the design object [25]. Scene manipulation can also be described as the way a user browses through and moves around in the virtual environment and defines the scale (or reference framework within which they move), in order to change their viewpoint. In the end, the key features of IVRIE, including being able to have various viewpoints and change the scale, along with the sense of navigation and direct interaction with virtual objects, allow the user's spatial presence and perception to reach higher levels, shaping a completely different user experience from what is found with other media.

### 2.2. Virtual Reality, Visualization and Spatial Design

Humans possess a suite of perceptual systems that allow them to sense their environment. Among the various senses, sight is the dominant component of human sensory perception [34]. Visual language is the basis and root of design creation. It is structured around the formation of design elements and rules for putting them together [35]. Designers rely heavily on visual language to communicate design ideas. The rapid technological advancements and decreases in the cost of VR have made it an affordable visualization/modeling tool [11].

The application of VR in the architectural design field has increased in recent decades; still, there is a lack of research clarifying VR's impacts as a design and visualization tool. The use of 3D VR visualization in architectural design deals mainly with volume conceptualization; it provides spatial information through proper interfaces to simulate depth, one of the essential components of spatial cognition [7,18].

Evidence suggests that the information representation improvements offered by VR lead to more qualitative design representations, including the representation of spaces from the users' perspective through full-scale 3D spatial information and the illusion of depth and immersion [23]. Since virtual worlds are not tied to physical reality, any information and specific complex data that can be visualized can also be made into a virtual 3D interpretation environment that a user can experience [27,36–38].

Conversely, some studies have criticized the functionality and use of VR in the design and construction fields, defining it as a passive tool that provides excellent visual feedback but is unable to inform regarding a design's problems and weak points. The majority of such research evaluated VR as a tool for passively viewing designs [2,6,39].

### 2.3. Virtual Reality, Design Thinking, and Design Approaches

Its spatial nature, scalability, immersion, and interaction make VR an explorable environment, enhancing visual thinking and playing a functional role in problem solving and inspiring design solutions [3,34]. Evidence suggests that VR enhances users' design thinking and enables them to produce more thoughtful and rational designs. Because of this enhancement, users are more aware of the 3D character of their designs, and their ability to visualize their designs improves; thus, they have the confidence to alter their approach to the design [6,18]. Improvements in spatial awareness resulting from VR have yielded faster development of users' ability to visualize spaces accurately and be more aware of design decisions' spatial impacts [5,7,10,23]. This improved spatial awareness affects the user's ability to recognize design problems and propose solutions throughout the design review process. In such a process, a design solution is evaluated for any possible failures concerning the program, function of spaces, and overall performance of the proposed design. As a representation, VR enables the reviewer to understand the proposed design solution and allows for a meaningful critique, facilitating their ability to overcome cognitive limitations and provide better spatial perception through the design's representation [4,5,7,23]. Immersive Virtual Reality's (IVR) ability to transfer spatial data to users, including a sense of scale and dimensions, and allowing for engagement and interaction with designed spaces on a human scale improve users' perception of the spaces

and affect their design thinking. VR adds the dimensions of immersion and interactivity to 3D computer-generated models and allows a kind of exploration not possible with other forms of representation [2,8,24].

The integration of IVR with other technologies (such as GIS) has been identified as a factor effective for finding design solutions and interpreting complex environmental situations [3]. Systems merged with VR are defined as decision support systems (DSS). Such software systems integrate tools that can manage the flow of information and work, guiding designers and planners in interpreting complex environmental situations and developing design solutions [39]. A successful DSS integrates visualization, predictive modeling, and communication techniques into an environment that immerses planners in the planning situation and designers in problem solving. Thus, such a set of tools enables the exploration of future scenarios from both the planning and design perspectives and displays them in various flexible, informative, and interactive formats [18,34,39,40].

## 3. Research Design and Methodology

The methodological framework of this study applies the recommended steps of explanatory sequential mixed methods research, which is a two-phase research design. The results of collected and analyzed quantitative data in the first phase are used to plan and build the second qualitative phase [41,42]. In this study's mixed sequential design, the whole process of conducting the experiments and gathering data in the first phase follows quantitative, comparative, within-subjects experiment design rules, and inferential statistical testing are used to analyze the gathered data. In the second phase, qualitative data, collected through questionnaires, are compared, analyzed, and interpreted concerning the results from quantitative data in the first phase. Data collection relies on two separate categories; first, measuring the volume/area of each participant's design results (quantitative data), and second, gathering Spatial Perception Questionnaire (SPQ). The proposed method for qualitative data collection is based on previous research on perception and sense of presence in VR environments, which adopted spatial presence or perception questionnaires to assess users' experiences in immersive environments [15,19,43]. The method for quantitative data collection is based on the authors' innovation for collecting precise data from the participants' real design performance and measurable design results. The source of the data generation is the experiments, which were designed to investigate how users understand a design's spatial structure and how two different virtual environments, IVRIE versus DT, impact that understanding based on variations in experiential/spatial factors (e.g., spatial perception, spatial presence, etc.).

### 3.1. Experiment Design

The experiment design of this study is based on a conceptual model for the experiment design, in which a tool/environment usage test would be conducted to compare the capability of transferring spatial data of a tool/environment to users. In addition, the differences in users' spatial decision making and performance based on this spatial data transmission was recorded. The experiment design's conceptual model, comprising three stages, is proposed by distinguishing five different key features of IVRIE from those of DT system in the first stage. The authors gathered these features from the literature review and pilot study results. As a design tool or design environment, IVRIE has five features important to this study: (a) full immersion, (b) a human scale and all other standard scales (e.g., 1:10, 1:100), (c) 360° viewshed without the need for a controller (by rotating the head, users can obtain an immersive view of their surroundings), (d) different and direct interactions with internal virtual design objects, and I the ability to move through the design. Different interactions mean the ability to interact with design objects by a mouse and one hand in DT system versus direct interaction in IVRIE using two hands and two controllers (or a pair of smart gloves). Other features such as immediate and real-time feedback have been considered common to both environments and thus are not considered here.

In the second stage of the conceptual model, user's experience/perception tested by applying different experiential/spatial factors and having users complete designated design tasks using either IVRIE or DT system. The experiential/spatial factors are based on the authors' hypotheses and the spatial experiences each virtual system offers. These factors are divided into four spatial branches as follows: (a) presence, (b) perception, (c) awareness, and (d) cognition. Combining these factors as components of users' experience/perception of spatial characteristics in design produces their learning/performance experience using each virtual environment, which is predicted to be different between IVRIE and DT system. Analyzing differences in the learning experience, such as how users' perceptions of design elements (e.g., scale/volume/depth) are different in IVRIE as compared to DT system, clarifies the impacts of each virtual environment on design learning outcomes (i.e., the third segment). The predicted differences in design elements perception, learning, and user performance are divided into four categories in order to answer the following questions: (a) Are there any differences between the perception of design elements in IVREA and DT system? (b) Are users' spatial design decisions different using IVRIE and DT system? (c) Are the spaces designed using IVRIE and DT system as a design environment/tool different in size and volume? (d) Do users reach different design solutions utilizing different virtual environments? Figure 1 demonstrates the flow of three parts of the conceptual experiment model and their relationships.

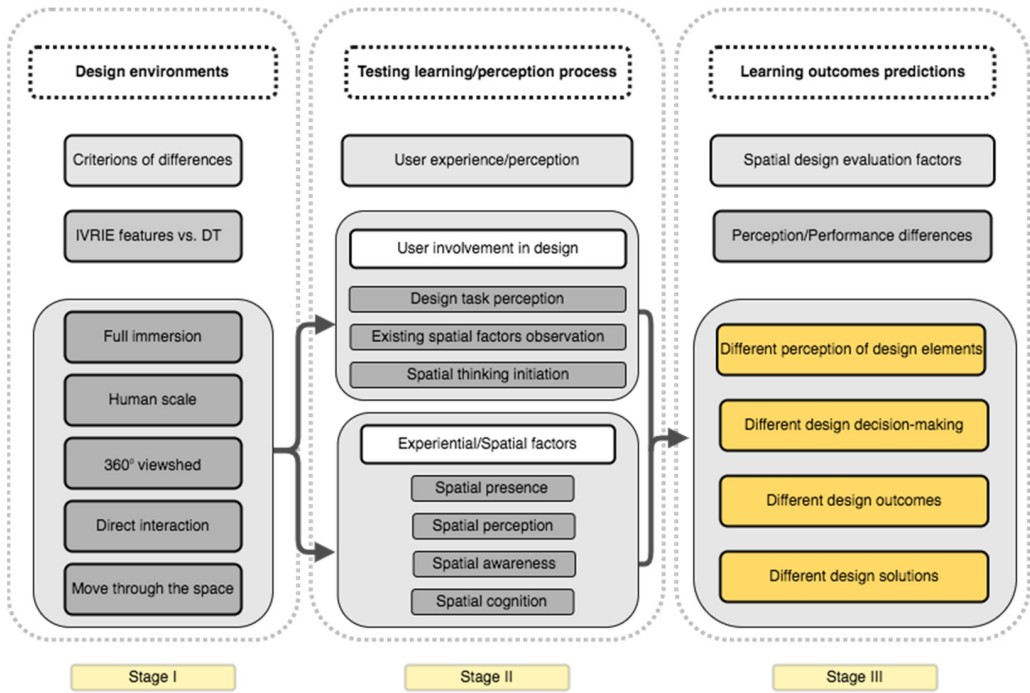

**Figure 1.** Proposed conceptual model for the experiment design.

Based on the conceptual model for the experiment design, the experiments of this study were conducted in six steps. Figure 2 illustrates the steps and consequences of the experiment plan.

In Step 1 of the experiment preparation process, virtual systems were selected. The SketchUp® software as the provider of semi-immersive virtual environment was selected for data collection of desktop-based VR part of the experiment, and the VR Sketch® program was chosen as IVRIE to collect data from participants' design results while they were working in a fully immersive interactive virtual environment. For the semi-immersive VR environment (DT system), this study used a conventional workstation consisting of a high-performance computer, 40" LCD monitor (and a regular size monitor), keyboard, and mouse (with the latter two serving as interaction devices). For the IVRIE, the study

employed the same high-performance computer and Oculus shift immersive VR system, including headset sensors, an Oculus headset, and two controllers serving as interaction devices. Both systems' setup and operating schema led participants to perform the same design tasks, while their engagement with virtual models occurred in different testing conditions, including different senses of immersion and interaction, equipment interfaces, and spatial presence. Figure 3 illustrates the setup and operating schema for both systems.

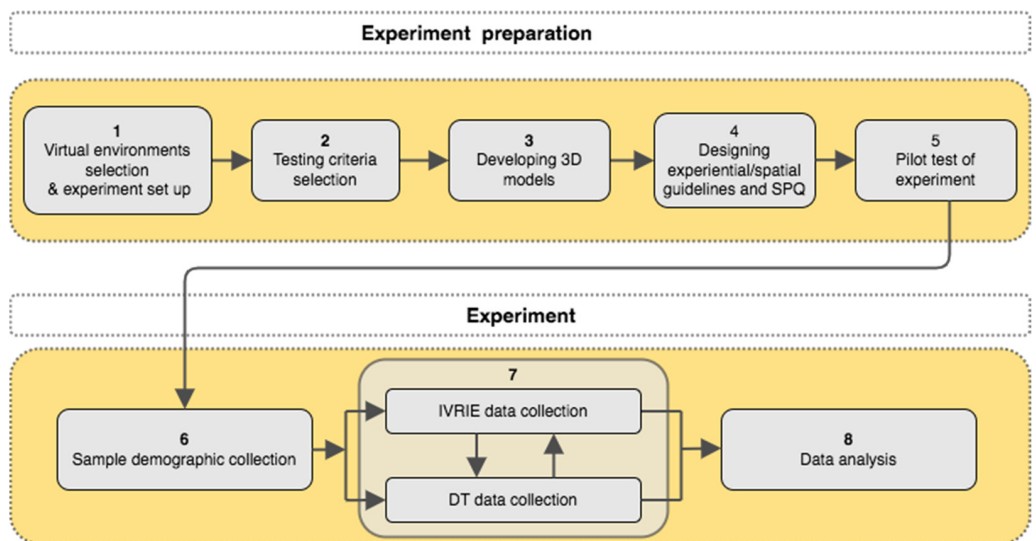

**Figure 2.** Flow chart of the experiment design.

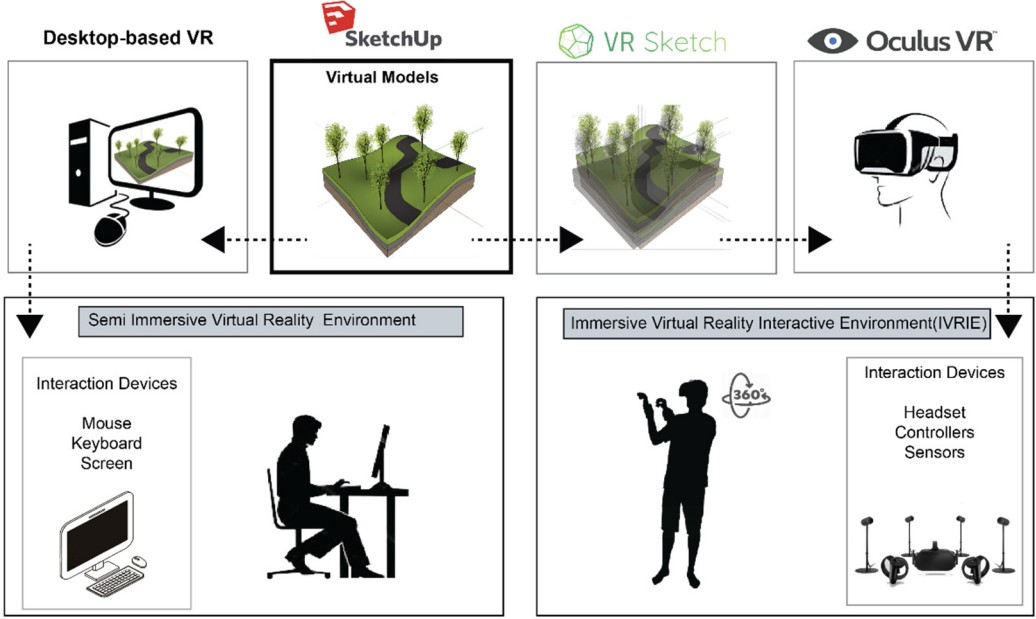

**Figure 3.** Setup and operating schema for the VR environments.

In Step 2, the criteria of the characteristics and specifications of virtual models were investigated and identified to be aligned with all the required quantitative data that authors aimed to extract from each participant according to their performance in each system. These criteria comprise the initial form and volume of the virtual models, the number of models in each scenario, the number of scenarios based on the differences in texture, and the sequence of scenarios and existing spaces in each of them.

In Step 3, the virtual models were developed with SketchUp®; for the IVRIE part of the experiment, the models were sent to the VR Sketch® extension. The customized VR Sketch

version allowed users to view and edit directly in VR and easily interact with design objects. The link between the SketchUp files and VR Sketch was immediate, and models could be seen simultaneously on the screen and through the headset. This characteristic of VR Sketch created a unique opportunity. While participants wore the headset and felt themselves entirely in an immersive virtual world, their actions (such as how they moved, interacted with objects, and even changed their viewpoint) could be seen and tracked by others (i.e., the experiment conductors) on the flat screen. The developed 3D models that participants worked with in both desktop and IVRIE systems consist of two sets of models, and each set comprised four different spaces (two corridors and two enclosure spaces). Each set of models had different textures in the inner surfaces of the spaces. In one set of the models, the inner surfaces of the spaces are covered with plain white color (which can be called devoid of texture), and the other one has patterned cover in inner surfaces (brick-pattern texture). The reason for having two sets of models with different textures was to obtain comprehensive data regarding spatial perception and design decisions as participants made choices in changing the volume and size to redesign the spaces when using the two different systems. Indeed, the impacts of full immersion and direct interaction with design objects with different specifications (e.g., texture: plain vs. patterned; form: corridor vs. enclosed) on spatial perception were evaluated. Then, the results of the designs generated in the IVRIE system were compared to those generated in the DT system. Additionally, the results generated within each system and between the scenarios, based on plain and patterned texture, were compared too. Figure 4 displays the bird's eye view of developed models to test volume/scale perception.

In Step 4, the spatial/experiential guidelines and SPQ were administered. Four different spatial/experiential guidelines were given to participants, each belonging to one of the virtual spaces in each scenario. To determine the effectiveness of both immersive and semi-immersive environments, spatial adjectives and experiential guidelines for manipulating and resizing the given spaces focused on volume/scale and distance/scale perceptions. Participants completed these design tasks in both systems and then responded to the questionnaire. Participants were allowed only to use the selection, push, and pull (move) commands to interact with virtual models in both systems to complete design tasks. The goal was for the participants to change, responding to spatial guidelines, the area and volume of the given spaces, and distances between design objects. Each participant's design results from both systems were saved and stored for later calculation and comparison of the area, volume, and distance between objects of redesigned/recreated spaces.

The SPQ has subjective opinion questions and gathers participants' evaluations of the helpfulness degree of IVRIE features, including full immersion, direct interaction, access to eye-level view, systems usage sequence, and systems' efficiency for design. All the answers to the SPQ's questions were coded into numeric scales for use in statistical tests and interpretation of analyzed results of quantitative data derived from measurements of resulting designs. In Step 5, a pilot experiment was conducted to test the equipment and perform final adjustments. In Step 6, sample demographic was collected. For this study, the sample consisted of 60 participants divided into two groups with different systems usage sequences.

Step 7 was data collection, and it was conducted one participant at a time. In the first group, each participant first worked on 3D virtual models in the IVRIE system and then with the desktop-based VR system (DT system). For the second group, this process was reversed. The value of group division is in having more data to compare and interpret concerning how participants' performance and design decisions differ by using virtual environments in different sequences. The goal was to determine if users had the same experience doing the same design task and familiarizing themselves with 3D models in virtual environments with different features and how their design decisions might differ.

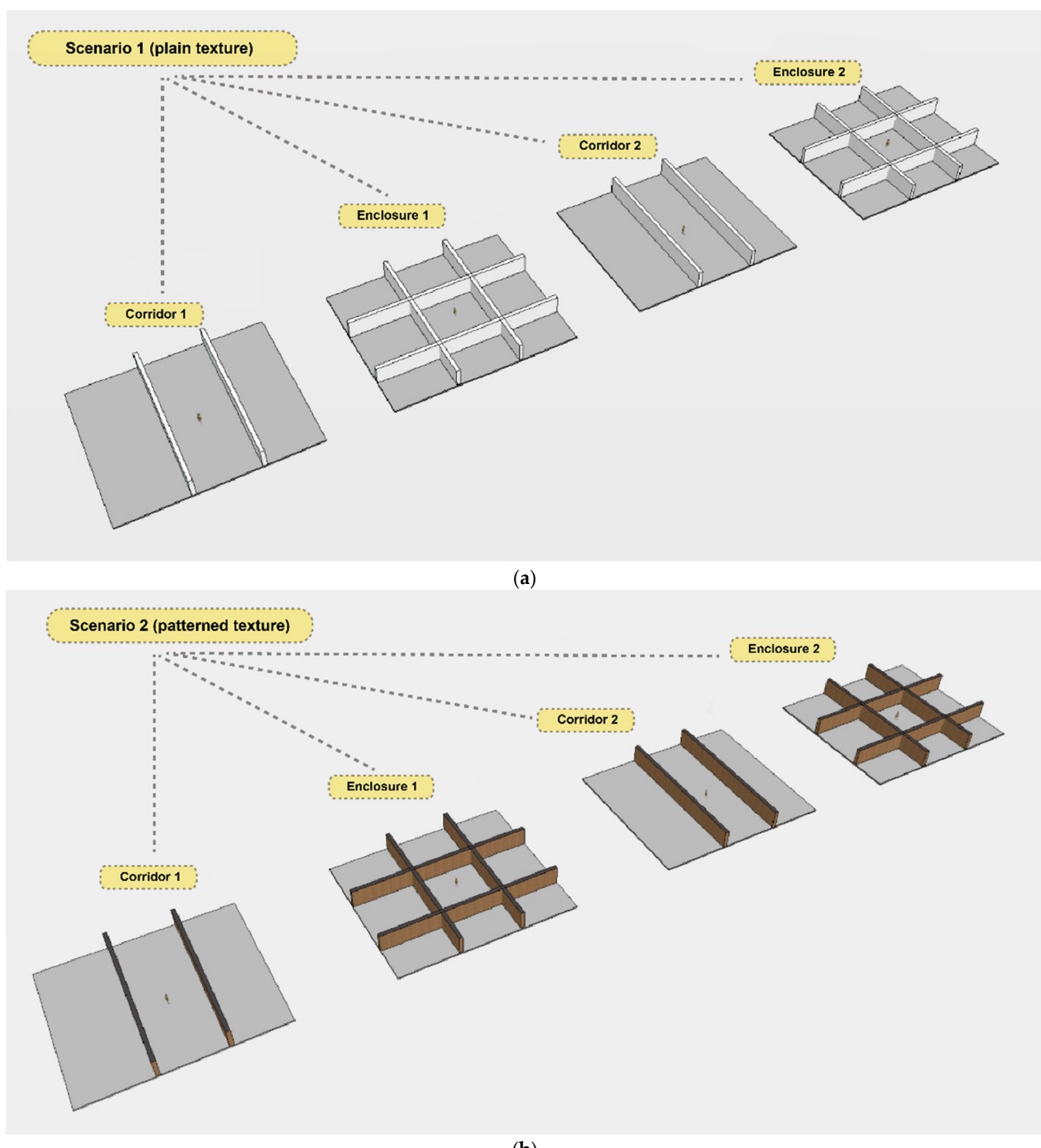

**Figure 4.** Models developed for the experiments: (**a**) scenario 1: plain texture; (**b**) scenario 2: patterned texture.

In Step 8, various statistical tests were performed. Some tests statistically examined the datasets individually, while others examined the data based on the relationships and connections among the datasets. The analysis consisted of two separate branches comprising analyses of selected tests to evaluate significant differences in the size and scale variations in the two systems and analyses of the first branch's results concerning categorized qualitative data collected from sample demographic.

*3.2. Data Collection and Analysis*

3.2.1. Descriptive Statistical Tests

In this research, the procedure for testing the hypotheses relied on statistical tests on the collected quantitative data. The results of the tests were analyzed and compared to collected qualitative data to facilitate the evaluation of the study's hypotheses and related objectives and questions. The quantitative data were extracted by measuring the size and scale of the spaces that each participant designed in each system, and various two-sample *t*-tests were performed to analyze these data. In all tests, the calculation of the *p*-value (which provides a statistical evaluation of two population means, averages, or percentages) validated the claim that there were variations in the size and scale of the spaces designed in each system based on participants' spatial perception and cognition. In other words, the *p*-value was used as a determinant of statistical significance, indicating whether differences in space size based on data collected under different conditions were significant. The level of significance adopted for all tests was 0.05; this produced a 95% confidence interval for determining the significance of differences between groups within the sample. Therefore, if the calculated *p*-value for each statistical test was lower than 5%, we concluded that the difference between the means of the two samples was significant. Thus, the results for the populations were not attributed to chance and could be assumed to be truthful.

3.2.2. Sample Profile and Data Diversity

The sample size in this research study was 60 participants, comprising design students and professionals in architecture, landscape architecture, industrial design, and graphic design. All the participants had familiarity with 3D models in desktop systems, and also, all of them had used VR headsets at least once before and experienced the sense of full immersion in some activities such as gaming. The data from each participant were collected via two methods and categorized into two corresponding categories. Each data category was analyzed individually and in connection to the other categories. The categories of collected data were as follows: (1) Obtained data from measuring the spaces designed by participants in the DT system and IVRIE system. The total data for each participant in this category was 16 numeric values from the measurements of 16 spaces (eight spaces in DT and eight spaces in IVRIE). (2) Data extracted from participants' answers to the spatial perception questionnaire (SPQ) questions. The total number of questions on the SPQ was four, producing four values. The sample was divided into two equal groups (each comprised 30 participants) based on each group's systems usage sequence. Half of the participants started the experiment with the DT system and, after completing the design tasks in that system, moved to IVRIE and finished the experiment there. The other half did the same but in the opposite direction, with the sequence being IVRIE first and DT second. During the process of running the experiments and collecting the data, instruction was offered to minimize the impacts of spatial memory on participants' decision making. Based on the instruction, the sequence of introducing the virtual spaces and giving the related spatial/experiential guidelines was different in each scenario. For all the participants, either start the experiment with IVRIE system or DT system, the sequence of facing a space for the first time and listening/reading the related guideline for redesigning that space started from space 1/guideline 1 and continued to space 4/guideline 4 in the plain scenario and in the textured scenario the sequence was 3, 2, 4 and 1. The same sequence is repeated when the participant has completed the experiment in the first system and moves to the second system. This variation in space/guideline sequence distracted participants' spatial memory at an acceptable level due to the time interval and variety of other spaces they experienced between facing spaces in the first scenario and their pairs (which were similar) in the second scenario within or between the systems.

3.2.3. Data Refinement

As mentioned before, the quantitative data collected for the analysis of this study were extracted from the space sizes each participant redesigned in the two systems, based on

given spatial/experiential guidelines. In each system (DT and IVRIE), 2 scenarios that included 8 spaces were available for each participant; thus, the total number of spaces available was 16, producing 16 values for each participant. Before using any values in the analysis, all were tested through interquartile ranges and verified as valid data (and not outliers). Outliers can make descriptive statistics such as mean, median, and range values misleading; thus, identifying them is essential for accurate statistical inferences.

Identification and removal of outliers (i.e., bad data) from the data collected in this study were based on two logical and functional rules utilized for all data collected from all participants. These rules were as follows. (1) If a participant had more than three outliers in the collected values (i.e., the numerical values extracted from the measurement of the redesigned spaces), then that participant and all his/her values (or data) were removed from the sample. (2) If a participant had three or fewer outliers, all outliers and their paired values were removed. In other words, if we removed an outlier that was a value from the measurement of a space in DT, then its paired value, the measurement of the same space in IVRIE, was removed, even if the paired value was not an outlier. This process was repeated by removing the outlier in one scenario and its pair value in the other for logically conducting comparative analyses between the systems. Since all the statistical tests in this branch of analysis (i.e., the two-sample *t*-test) relied on the comparability of values and their pairs, this data refinement prevented the test results from being inaccurate because of good and bad data combinations.

The numerical values for each participant were divided into two categories. This division was based on the difference in units of measurement. Each participant worked on eight corridor spaces in both systems, four in DT and four in IVRIE. The numerical values extracted from the corridor spaces were the distances between the two parallel walls (i.e., the width). The measurement unit for these spaces was feet. The same participant worked on eight enclosure (room-shaped) spaces in both systems (four in DT and four in IVRIE). The values extracted from the enclosure spaces were the inner areas of the spaces in square feet. Thus, the data collected from each participant included eight values in feet and eight values in square feet for both systems. The initial width (the distance between the two parallel walls) of all corridor-form spaces was 10 ft. The initial area of all enclosed-form spaces was 10 by 10 ft. (100 sq. ft.).

## 4. Results

The results and analysis of this study are categorized into four branches. The first three branches comprise the statistical analysis, in which the role of texture, spatial/experiential guidelines' utilization, and systems usage sequence in affecting the spatial perception of users and resulting in space size variations within and between the IVRIE and DT systems are analyzed. The latter branch, comprising four sub-branches, is statistical analyses comparing and relating the results of the three first branches based on sample population divisions regarding participants' evaluations of virtual systems' features and efficiency.

### 4.1. Texture and Comparison of Space Size Variations within and between Systems

The first branch of analysis is statistical comparisons of space size variations between and within systems based on the role of texture. The testing conditions for between systems comparison included the following: (1) similar spaces, (2) similar texture, (3) common guidelines, and (4) different systems. The conditions of within systems comparison were (1) similar spaces, (2) common guidelines, (3) different textures, and (4) similar systems. The results of the tests and comparisons for this analysis are outlined below.

4.1.1. Between-Systems Comparison

In this portion of the analysis, both space categories (i.e., enclosure and corridor) in each scenario (plain and patterned) were compared between the two systems. The statistical analysis identified the possible role of texture in the significant space size differences found

between the two systems in terms of similar spaces with similar textures. The results of the tests and comparisons are as follows.

Texture: Plain; System: IVRIE vs. DT

This analysis found that the average widths (i.e., the distance between two parallel walls) of corridor spaces and the average areas of enclosure spaces with plain texture were not significantly different between the two systems. Calculated *p*-values (corridors (0.83, non-significant), enclosures (0.13, non-significant)) indicated that between the DT and IVRIE systems, participants' spatial decision making for similar spaces with plain texture did not result in significant space size differences.

Texture: Patterned; System: IVRIE vs. DT

The analysis indicated that the average widths of corridor spaces and the average areas of enclosure spaces with patterned texture were significantly different between the two systems. Calculated *p*-values (corridors (0.003, significant), enclosures (0.003, significant)) indicated that between the two systems, participants' spatial decision making for similar spaces with patterned texture (brick-pattern texture) did not result in significant space size differences. Table 1 presents the size variations among the spaces between systems.

**Table 1.** Size variations for spaces within scenarios and between systems.

| Within Scenarios/Between Systems | | | | | | | | | | | |
|---|---|---|---|---|---|---|---|---|---|---|---|
| **Scenarios** | | **Plain** | | | | | **Patterned** | | | | |
| **Systems** | | **DT** | | **IVRIE** | | | **DT** | | **IVRIE** | | |
| | | **Mean** | **ST** | **Mean** | **ST** | ***p*-Value** | **Mean** | **ST** | **Mean** | **ST** | ***p*-Value** |
| Space categories | Corridor (ft.) | 14 | 5.25 | 13.9 | 5.02 | 0.83 | 15.6 | 5.79 | 13.8 | 5.23 | 0.003 * |
| | Enclosure (ft$^2$) | 867.4 | 484.6 | 951.5 | 389.7 | 0.13 | 958.2 | 600.4 | 792.1 | 327.1 | 0.003 * |

The (*) symbol indicates significant *p*-value.

### 4.1.2. Within-System Comparison

In this portion of the research, both space categories (i.e., enclosure and corridor) in each system were compared with their paired spaces with different texture. The statistical analysis identified the possible role of texture in the resulting significant average space size differences within each system in terms of similar spaces with different textures. The results of the tests and comparisons are outlined below.

System: DT; Texture: Plain vs. Patterned

A comparison of the average sizes (widths) of corridor spaces and the average areas of the enclosure spaces with plain texture were significantly different from the average sizes of corridors and enclosure spaces with patterned texture in the DT system. Calculated *p*-values (corridors (0.02, significant), enclosures (0.0002, significant)) indicated that in DT system, participants' spatial decision making for similar spaces with different texture resulted in significant space size differences.

System: IVRIE; Texture: Plain vs. Patterned

A comparison of the average sizes (widths) of corridor spaces and the average areas of the enclosure spaces with plain texture were not significantly different from the average sizes of corridors and enclosure spaces with patterned texture in the IVRIE system. Calculated *p*-values (corridors (0.22, non-significant), enclosures (0.09, non-significant)) indicated that in the IVRIE system, the presence of texture did not affect participants' spatial decision making for similar spaces and resulted in insignificant space size variations. Table 2 presents the size variations among the spaces within systems.

**Table 2.** Size variations for spaces between scenarios and within systems.

| Between Scenarios/Within Systems | | | | | | | | | | | |
|---|---|---|---|---|---|---|---|---|---|---|---|
| Systems | | DT | | | | | IVRIE | | | | |
| Scenarios | | Plain | | Patterned | | | Plain | | Patterned | | |
| | | Mean | ST | Mean | ST | *p*-Value | Mean | ST | Mean | ST | *p*-Value |
| Space categories | Corridor (ft.) | 14.4 | 5.19 | 15.3 | 5.88 | 0.02 * | 14 | 5.06 | 13.6 | 5.2 | 0.22 |
| | Enclosure (ft$^2$) | 895.2 | 503.2 | 1074.4 | 691.6 | 0.0002 * | 743.4 | 392.8 | 790.2 | 327.01 | 0.09 |

The (*) symbol indicates significant *p*-value.

### 4.2. Spatial/Experiential Guidelines and Comparison of Space Size Variations within and between Systems

This portion of the research examined size differences between and within systems based on the role of given experiential/spatial guidelines. Four different experiential/spatial guidelines, each belonging to redesigning a space in a scenario, were given to participants. A participant used each guideline two times for redesigning a space, once in IVRIE and once for that space's pair in the DT system. For example, the guideline, "Design space number 2 to be a comfortable space for two people to gather", was used for one of the enclosure spaces with plain texture, once in IVRIE and once in the DT system. The same guideline was used for the same space while it had patterned texture in both systems.

The comparisons for eight different spaces in each system, divided into two scenarios (plain and patterned), were performed between and within systems. The conditions for between systems comparison included: (1) similar spaces, (2) common guidelines, (3) similar texture, and (4) different systems. The conditions of within systems comparison were (1) similar spaces, (2) common guidelines, (3) similar systems, and (4) different textures. The results of the tests and comparisons for this analysis are outlined below.

#### 4.2.1. Between-System Comparison

This portion of the research compared the sizes of all four spaces in each scenario in the desktop-based VR system with their paired spaces in the same scenario in the IVRIE system under a common experiential/spatial guideline. The results of the tests and comparisons are outlined below.

#### Texture: Plain: System: IVRIE vs. DT

Comparisons revealed that the sizes (widths) of corridor spaces with plain texture between two systems and with the common experiential/spatial guideline focusing on designing the spaces for one and three persons to walk in were not significantly different. Calculated *p*-values: ((corridor 1: one person (0.13, non-significant)), (corridor 2: three persons (0.092, non-significant)) indicated that participants' spatial decisions for the corridor spaces produced under a common guideline did not differ significantly between the two systems.

For enclosure spaces, comparisons between the two systems indicated significant difference in the inner areas of these two spaces while both have plain texture and redesigned by a common experiential/spatial guideline focusing on designing the spaces for two and ten persons to gather. Calculated *p*-values: (enclosure 1: two persons (0.011, significant)), (enclosure 2: ten persons (0.023, significant)) showed that the spatial decisions of the participants for the enclosure spaces produced under a common guideline resulted in a significant difference in the inner areas between the two systems.

#### Texture: Patterned; System: IVRIE vs. DT

Comparisons revealed that the sizes (widths) of corridor spaces with patterned texture between two systems and with the common experiential/spatial guideline focusing on designing the spaces for one person to walk in was significantly different but in contrast

with designing for three persons. Calculated *p*-values: ((corridor 1: one person (0.008, significant)), (corridor 2: three persons (0.5, non-significant)) indicated that participants' spatial decisions for the corridor spaces with patterned texture resulted in significant size differences when the guidelines lead users to decide about the width of corridors for more than one person to be in.

For enclosure spaces, comparisons between the two systems indicated significant difference in the inner areas of these two spaces while both have plain texture and redesigned by a common experiential/spatial guideline focusing on designing the spaces for two and ten persons to gather. Calculated *p*-values: ((enclosure 1: two persons (0.001, significant)), (enclosure 2: ten persons (0.0006, significant)) showed that the spatial decisions of the participants for the patterned enclosure spaces produced under a common guideline resulted in a significant difference in the inner areas between the two systems.

### 4.2.2. Within-System Comparison

This section of the research compared the sizes of all four spaces in Scenario 1 (plain) with their paired spaces in Scenario 2 (patterned) within each system and under a common experiential/spatial guideline. The results of the tests and comparisons are outlined below.

### System: DT; Texture: Plain vs. Patterned

Comparisons revealed that the sizes (widths) of corridor spaces with different textures in the DT system and with the common experiential/spatial guideline focusing on designing the spaces for one and three persons to walk in were significantly different. Calculated *p*-values: ((corridor 1: one person (0.02, significant)), (corridor 2: three persons (0.000, significant)) indicated that participants' spatial decisions for the corridor spaces with different texture and common guidelines resulted in significant size differences within this system.

For enclosure spaces, comparisons between similar spaces with different texture within the DT system indicated significant difference in the inner area of the enclosure space, designated to have the capacity for ten persons but in contrast with the one, with capacity of two persons to gather. Calculated *p*-values: ((enclosure 1: two persons (0.09, non-significant)), (enclosure 2: ten persons (0.001, significant)) showed that the spatial decisions of the participants for the similar enclosure spaces when texture differs and guidelines lead users to decide about the size of space for more than two persons to be in resulted in a significant difference in the inner areas within this system.

### System: IVRIE; Texture: plain vs. patterned

Comparisons revealed that the sizes (widths) of corridor spaces with different textures in the IVRIE system and with the common experiential/spatial guideline focusing on designing the spaces for one and three persons to walk in were significantly different. Calculated *p*-values: ((corridor 1: one person (0.02, significant)), (corridor 2: three persons (0.000, significant)) indicated that participants' spatial decisions for the corridor spaces with different texture and common guidelines resulted in significant size differences within this system.

For enclosure spaces, comparisons between similar spaces with different texture within IVRIE focusing on designing the spaces for two and ten persons to gather in were not significantly different. Calculated *p*-values: ((enclosure 1: two persons (0.45, non-significant)), (enclosure 2: ten persons (0.67, non-significant)) showed that the spatial decisions of the participants for the similar enclosure spaces when texture differs and guidelines lead users to decide about the size of space for two or more persons to be in resulted in a non-significant difference in the inner areas within this system. Table 3 presents the summary of the size variations of each space between systems/within scenarios and within systems/between scenarios.

**Table 3.** Size variations of each space between systems/within scenarios and within systems/between scenarios.

| | | Between Systems Comparison DT vs. IVRIE | | | | | | Within Systems Comparison SC1 vs. SC2 | | | | | |
|---|---|---|---|---|---|---|---|---|---|---|---|---|---|
| | | SC1 (Plain) | | | SC2 (Patterned) | | | DT | | | IVRIE | | |
| | | Mean | | *p*-Value | Mean | | *p*-Value | Mean | | *p*-Value | Mean | | *p*-Value |
| Space Categories | | DT | IVRIE | | DT | IVRIE | | SC1 | SC2 | | SC1 | SC2 | |
| Space 1 | Corridor 1 | 9.1 | 10.1 | 0.1 | 10.13 | 9 | 0.008 * | 9.1 | 10.13 | 0.02 * | 10.18 | 9 | 0.02 * |
| Space 2 | Enclosure 1 | 391.1 | 446.8 | 0.01 * | 539.2 | 522.3 | 0.0019 * | 397.1 | 522.6 | 0.09 | 446.6 | 524.2 | 0.0006 * |
| Space 3 | Corridor 2 | 19.6 | 17.9 | 0.09 | 21.3 | 18.7 | 0.5 | 19.9 | 20.8 | 0.00008 * | 18.1 | 18.57 | 0.4 |
| Space 4 | Enclosure 2 | 1241.4 | 1081 | 0.02 * | 1316.5 | 1046.9 | 0.0006 * | 12.74.1 | 1465.8 | 0.001 * | 1029.7 | 1046.9 | 0.6 |

The (*) symbol indicates significant *p*-value.

*4.3. System Usage Sequence and Comparison of Space Size Variations between Systems*

In this analysis, the sample was divided into groups with equal numbers of participants. The first group consisted of 30 participants who started the experiment with a desktop-based VR system and then worked in IVRIE as the second system. The other group, consisting of the same number of participants, completed the experiment in the opposite order. The test compared the average sizes of enclosure and corridor spaces for the two groups between the two systems. The comparison condition for this analysis was identifying the system usage sequence as an active factor and ignoring other factors such as texture and spatial/experiential guideline utilization. The statistical analysis revealed the following results, comparing space size for each category.

4.3.1. Between-System Comparison

This research compared the average sizes of all spaces in each space category (all enclosure and corridor spaces in the plain and patterned scenarios). The goal of this comparison was to identify differences in average space size for each space category and its pair, as recorded when participants were becoming familiar with the spaces and working either first with DT and then IVRIE or vice versa. This statistical test identified the effects of system usage sequence on the resulting differences in average space size between the systems and for each group. The results of the test are as follows.

Sequence: DT to IVRIE

A comparison of the average sizes (widths) of all corridor spaces for the two systems for the first group of participants (who started the experiment with the DT system and completed it with IVRIE) showed significant differences and not significant for the inner area of enclosure spaces.

The calculated *p*-values (corridors (0.03, significant), enclosures (0.34, non-significant)) verified that participants' spatial decisions for corridor spaces in IVRIE with a backup of spatial cognition for these spaces from the DT system resulted in significant differences in widths. For enclosure spaces, this system usage sequence did not have any effects on spatial decision making for similar spaces between the two systems.

Sequence: IVRIE to DT

A comparison of the average widths of the corridor spaces between the two systems for participants beginning the experiment with the IVRIE system and completed it with DT indicated that the sizes were not significantly different but in contrast with the inner area of enclosure spaces, designed with this systems sequence.

The calculated *p*-values (corridors (0.1, non-significant), enclosures (0.004, significant)) verified that participants' spatial decisions for corridor spaces in DT with a backup of spatial cognition for these spaces from IVRIE system did not result in significant differences in widths. In contrast, the inner area of enclosure spaces, designed in the two systems with

this system usage sequence, were significantly different. Table 4 presents the summary of the size variations based on systems usage sequence for both groups between systems.

**Table 4.** Size variations based on systems usage sequence for both groups between systems.

| | Between Systems Comparison | |
| --- | --- | --- |
| | **Group 1** | **Group 2** |
| | Sequence | |
| | **DT: 1    IVRIE: 2** | **DT: 2    IVRIE: 1** |
| Space Categories | *p*-value | |
| Corridor (ft.) | 0.03 * | 0.1 |
| Enclosure (ft$^2$) | 0.3 | 0.004 * |

The (*) symbol indicates significant *p*-value.

### 4.4. System Features Perception and Comparison of Space Size Variations between Systems

In this branch of the study, the participants' perceptions and evaluations of systems' features' efficiency in transferring spatial data and affecting their spatial decision making concerning their actual design results are analyzed. The collected qualitative data for statistical analysis is participants' responses to the questions on SPQ, focusing on the usability and effectiveness of IVRIE's features compared to the DT system. In the research design, these features were hypothesized as the factors that could affect the perception of spatial factors of virtual spaces and enable users to feel the spatial sense of their designed spaces differently in IVRIE compared to the DT system.

SPQ collects participants' self-evaluations of the helpfulness levels of IVRIE's features, including immersion, direct interaction, and access to eye-level view in Likert-scale of "very helpful" to "not at all helpful". Additionally, one question on SPQ focuses on participants' perceived accuracy and usefulness of IVRIE compared to the DT system in allowing them to effectively create the intended spatial sense in virtual spaces.

Participants were allowed to choose one of the answer options for each question, designated to one of the IVRIE's features. Based on the number of participants, who chose an answer option to each question, the population percentage of that group was calculated and then related to existing quantitative data for that portion of the sample. The quantitative data for this analysis are the numeric values from the measurements of the width of corridors and area of enclosure spaces with patterned textures designed by participants in both IVRIE and DT systems.

Between both scenarios (plain and patterned) utilized in experiments, the patterned one is chosen based on revealing significant space size variations between two systems under testing the participants' spatial perception affected by the kind of texture and utilization of spatial/experiential guidelines. The calculated *p*-values for both space categories (corridor and enclosure) in the patterned scenario between two systems for the whole sample are 0.003, which presented significant space size variations (see Table 1). The results of the tests and comparisons of space size variations between the two systems in relation to participants' self-evaluation of systems' features' usefulness are outlined below.

#### 4.4.1. Direct Interaction with Design Objects and Space Size Variations between Systems

The qualitative data regarding the perceived helpfulness of 'direct interaction with design objects in understanding the spatial factors of spaces' is categorized by the percentage of the participants' self-evaluation choices. Of the sample population, 82% of participants chose "very helpful", 15% "somewhat helpful", 2% "slightly helpful", and 0% "not at all helpful." Comparing the average widths of the corridors and area of enclosure spaces designed by the group1 ("very helpful" answer choice) between the two systems indicated significant size differences for both space categories for this group.

The calculated *p*-values (corridors (0.014, significant), enclosures (0.019, significant)) for this group show that the perception of these participants (82% of the sample size) for the level of helpfulness of direct interaction with virtual objects in their spatial decision making and being aware of the role of this IVRIE feature in their spatial cognition is aligned with their real design results and size variations of produced spaces in the two systems. On average, design results of this group have a higher mean value in DT system compared to IVRIE for the width and area of both space categories. The comparison of the size variation of designed spaces by this group in two systems with the 95% confidence interval for the corridors' width was between 1.9 and 2.9 feet (lower CI (95%) = 1.9, upper CI (95%) = 2.9) and for enclosures' area were between 135 and 313.5 square feet (lower CI (95%) = 135, upper CI (95%) = 313.5).

The comparison of size variations for the other groups (answer options of less than "very helpful") does not show significant differences in space sizes between the two systems. The calculated *p*-values for space size variations for these participants were non-significant, which clarifies the alignment of their awareness about the ineffectiveness of this feature in their spatial cognition and differing design results.

### 4.4.2. Spatial Cognition in Eye-Level View and Space Size Variations between Systems

The majority portion of the sample identified the necessity and helpfulness of having access to the eye-level view for understanding the spatial factors of design and making decisions about the feeling of spaces in IVRIE. Of the sample population, 93% of participants chose "very helpful", 5% "somewhat helpful", 2% "slightly helpful", and 0% "not at all helpful." Comparing the average widths of the corridors and area of enclosure spaces designed by the group1 ("very helpful" answer option) in two systems indicated significant size differences for both space categories for this group.

The calculated *p*-values (corridors (0.014, significant), enclosures (0.009, significant)) for group1 show that the perception of these participants (93% of sample size) for the level of helpfulness of seeing surroundings in eye-level view for spatial perception and decision making and being aware of this IVRIE feature is aligned with their real design results and size variations of produced spaces in the two systems. On average, design results of this group have a higher mean value in the DT system compared to IVRIE for the width and area of both space categories. The comparison of the size variation of spaces designed by this group in two systems with the 95% confidence interval for the corridors' width were between 2.1 and 3.1 feet (lower CI (95%) = 2.1, upper CI (95%) = 3.1) and for enclosures' area were between 140 and 298 square feet (lower CI (95%) = 140, upper CI (95%) = 298).

The comparison of size variations for other groups (participants with answer options of less than "very helpful") does not show any significant differences in space sizes between the two systems. The calculated *p*-values for space size variations for this portion of the sample were non-significant, which clarifies the alignment of the awareness of these participants in the uselessness of this kind of view or being dependent on it in their spatial cognition and production of different design results.

### 4.4.3. Sense of Full Immersion and Space Size Variations between Systems

Most of the participants distinguished the helpfulness of being fully immersed and capable of walking through the virtual spaces in cognition of spatial factors in design and making decisions about the feeling of spaces in IVRIE. Of the sample population, 82% of participants chose "very helpful", 18% "somewhat helpful", 0% "slightly helpful", and 0% "not at all helpful." Comparing the average widths of the corridors and area of enclosure spaces designed by the first group ("very helpful" answer option) in two systems indicated significant size differences for both space categories for this group. The calculated *p*-values (corridors (0.017, significant), enclosures (0.03, significant)) for group1 show that the perception of these participants (82% of sample size) for helpfulness level of full immersion sensing in spatial perception and decision making and awareness of this critical IVRIE feature is aligned with their real design results and size variations in

produced spaces in two systems. On average, the design results of this group have a higher mean value in DT system compared to IVRIE for the width and area of both space categories. The comparison of the size variation of spaces designed by this group in two systems with the 95% confidence interval for the corridors' width were between 2 and 3.5 feet (lower CI (95%) = 2, upper CI (95%) = 3.5) and for enclosures' area were between 126 and 299 square feet (lower CI (95%) = 126, upper CI (95%) = 299.9).

The comparison of size variations for other groups (participants with answer options of less than "very helpful") does not show significant differences in space sizes between the two systems. The calculated *p*-values for space size variations for this portion of the sample were non-significant, which clarifies the sense of full immersion in IVRIE and semi-immersion in DT system did not affect the size of their design results and is in alignment with their self-evaluation about having the similar perception of this feature in both systems.

### 4.4.4. Perceived Systems' Accuracy in Creating the Spatial Sense and Space Size Variations between Systems

The efficiency and usefulness of IVRIE compared to DT system in allowing the users to effectively create the intended spatial sense through designing the spaces most accurately were identified by the majority of the sample's population. Of the sample population, 80% of participants chose "IVRIE", 3% "desktop system", and 17% "both systems equally." Comparing the average widths of the corridors and area of enclosure spaces designed by the first group (IVRIE advocates) in two systems indicated significant size differences for both space categories for this group. The calculated *p*-values (corridors (0.024, significant), enclosures (0.006, significant)) for group1 show that the evaluation of these participants (80% of sample size) for efficiency of IVRIE in allowing users to effectively create the intended spatial sense in designing and spatial-decision making is completely aligned with their real design results and size variations of created/redesigned spaces in two systems. On average, design results of this group have a higher mean value in DT system compared to IVRIE for the width and area of both space categories. The comparison of the size variation of spaces designed by this group in the two systems, with the 95% confidence interval, was as follows: between 1.9 and 3.1 feet (lower CI (95%) = 1.9, upper CI (95%) = 3.1) for the corridors' width; between 132 and 309.5 square feet (lower CI (95%) = 132, upper CI (95%) = 309.5) for the enclosures' area.

The comparison of size variations for other groups (advocates of DT system or both systems equally) does not show significant differences in space sizes between the two systems. Although the design results of these two groups have a bit higher mean value in DT system than IVRIE for the width of corridors and area of enclosures, the calculated *p*-values for space size variations between the two systems were non-significant. The consistency of spatial decision making of these groups in two systems, which resulted in the similarity of size and volume of design results in both systems, aligns with their preference in using both systems equally or using the DT system solely for design. Table 5 presents the summary of systems' features statistical analysis.

**Table 5.** Space size variations and participants' system features perception.

| Perception Question | Focused Feature | Answer Option | Population Percentage | *p*-Value | | | |
|---|---|---|---|---|---|---|---|
| | | | | Corridor Spaces | CI (95%) Ft | Enclosure Spaces | CI (95%) Ft$^2$ |
| Q1 | Direct interaction | 1 | 82% | 0.014 * | (1.9, 2.9) | 0.019 * | (135, 313.5) |
| | | 2 | 15% | 0.51 | - | 0.2 | - |
| | | 3 | 3% | 0.3 | - | 0.9 | - |
| | | 4 | 0% | - | - | - | - |

**Table 5.** *Cont.*

| Perception Question | Focused Feature | Answer Option | Population Percentage | p-Value | | | |
|---|---|---|---|---|---|---|---|
| | | | | Corridor Spaces | CI (95%) Ft | Enclosure Spaces | CI (95%) Ft$^2$ |
| Q2 | Eye-level view | 1 | 93% | 0.014 * | (2.1, 3.1) | 0.009 * | (140, 298) |
| | | 2 | 5% | 0.8 | - | 0.7 | - |
| | | 3 | 2% | - | - | - | - |
| | | 4 | 0% | - | - | - | - |
| Q3 | Full immersion | 1 | 82% | 0.017 * | (2, 3.5) | 0.037 * | (126, 299) |
| | | 2 | 18% | 0.2 | - | 0.1 | - |
| | | 3 | 0% | - | - | - | - |
| | | 4 | 0% | - | - | - | - |
| Q4 | Perceived overall system accuracy | DT | 3% | 0.3 | - | 0.2 | - |
| | | IVRIE | 80% | 0.02 * | (1.9, 3.1) | 0.006 * | (132, 309.5) |
| | | Both systems | 17% | 0.3 | - | 0.6 | - |

Answer options: 1 = very helpful; 2 = somewhat helpful; 3 = slightly helpful; 4 = not at all helpful. The (*) symbol indicates significant *p*-value.

## 5. Discussion

This research attempted to identify how two different IVR systems can shape users' perception of spatial factors and impact their spatial decision making and performance within each system. The findings indicate that based on the characteristics of the IVRIE and DT systems, there is a difference in levels of spatial presence between the two systems, which can result in different design outcomes. Statistical analyses revealed that the size and volume of spaces designed by users are significantly different when utilizing the two different virtual environments. In addition, these differences may result from variations in spatial decision making within each system. On-going research efforts need to explore additional factors that may impact users' spatial thinking and decision making. Variables such as the degree of spatial presence, spatial memory, and experience with spatial design may be impacting spatial decisions made by the research subjects.

In the methodology and experiment design of this research, the assumption was that the spatial presence of users would be different between the IVRIE and DT systems and would result in different spatial decisions. Factors that were felt to have the potential to cause these differences were the sense of full immersion versus semi-immersion or the sense of direct interaction with virtual objects versus indirect interaction [15,44,45]. One of the limitations in the methodology for finding possible differences in spatial decision making of users within each system was controlling participants' distraction levels. When participants were completing the experiment in IVRIE using [46] a headset, they could still hear the voices and other sounds from their real-world context. Thus, part of their spatial presence in the virtual environment was affected or even dependent on the real world context of the experiment. Another limitation was the impact of spatial memory on spatial decision making. Although researchers took some actions to minimize the potential effect of users' spatial memory, such as changing the sequence of virtual models in each system, participants could still transfer some spatial familiarity between the two systems. Another factor that could have had an impact on experimental outcomes was the nature of the experiential/spatial guidelines that were used. Researchers attempted to provide simple and easily understandable guidelines for determining the scale and volume of spaces. These guidelines focus on creating spaces for a specific number of people to occupy or pass through the space "comfortably". It is understood that these qualitative decisions could be biased by different personal or cultural backgrounds. However, this type of qualitative descriptor is similar to those used by designers in the creation of spaces. One interesting finding that warrants further exploration is that many of the younger participants, following the same guidelines, designed more spacious enclosures than the older participants.

It is hoped that the findings of this study will encourage future research regarding the impact of the utilization of various VR systems on architectural design outcomes and education. Research on the effect on design education and design outcomes exploring how variations in the levels of immersion, types of interaction, and view usage patterns in virtual environments will be critical as we continue to incorporate this emerging technology into the processes that shape our physical environment.

## 6. Conclusions and Future Vision

The overarching goal of this research was to determine the differences in spatial perception, cognition, awareness, and decision making between DT systems and immersive IVRIE. The analyses of the research data were categorized into two main branches. The first was statistical analyses of quantitative data extracted from measurements of the sizes of virtual spaces designed by participants when using each system. These analyses considered the absence or presence of textures, application of descriptive spatial/experiential guidelines, and variations in system usage sequence. The second branch of the statistical analyses focused on qualitative data extracted from participants' evaluations of IVRIE, in relation to quantitative data comprising measurements of actual design results. The outcomes of the analyses are summarized below.

Branch 1: Findings of analyses of descriptive and inferential statistical testing of quantitative data:

- The differences between the IVRIE and DT systems in terms of providing spatial presence affected users' spatial perceptions and led them to different spatial decisions; the result was significant differences in the sizes of spaces designed in each system.
- Inferential statistical tests revealed that on average, the sizes of the designed spaces were smaller in IVRIE than in the DT system, and the consistency of space sizes was improved.
- The presence of textures impacted spatial perception and led users to make different spatial decisions between the IVRIE and DT systems, resulting in a significant space size variation. In addition, the inclusion of textures impacted users' spatial decisions when using the DT system but did not have a significant impact in IVRIE.
- System usage sequence impacted users' spatial perception differently based on the type of virtual space. When the system usage sequence was first IVRIE and then the DT system, users' spatial decision making resulted in significant size variations in enclosure spaces but not corridors. When the DT system was used first and then IVRIE, only the corridor spaces showed significant size variations.
- Use of a common spatial/experiential guideline for designing a space (once in IVRIE and once in the DT system) frequently resulted in significant space size variations. Additionally, when the texture was changed from plain to patterned, using a common spatial/experiential guideline for designing the space frequently resulted in significant size variations.

Branch 2: Findings of analyses of descriptive and inferential statistical testing of qualitative data:

- The features of IVRIE (i.e., direct interaction with design objects, browsing surroundings via an eye-level view, and a sense of full immersion) were recognized by 85% of participants as "very helpful" in spatial decision making. A comparison of the sizes and areas of similar spaces designed by participants who found those features "very helpful" showed significant differences between the two systems.
- A substantial majority of the participants felt that the IVRIE system was more accurate than the DT system, allowing them to more effectively create the intended spatial sense. Statistical comparisons of the spaces designed by participants who perceived IVRIE as the most accurate system (80% of the sample population) revealed that on average, the sizes of the spaces designed/created in IVRIE, regardless of form, volume, and texture, were smaller than their paired spaces designed in the DT system.

This research tested the differences in spatial perception and performance of users of two different virtual systems. The findings indicate that users' spatial decision making

was affected by the features of each system and resulted in significant size differences in the virtual spaces designed. However, numerous variables and factors still demand exploration. Aspects of spatial thinking, along with spatial memory and its role in spatial decision making, require further consideration.

The degree of spatial presence experienced in a virtual environment may vary both as a function of individual user differences and the characteristics of the virtual environment. The way in which a user constructs a spatial model may be related to the presence that user gains in the given virtual environment [11,19,22,46].

Spatial memory was another topic considered in this study, especially when participants worked on models in IVRIE. Although the sequence of spaces and guidelines for each scenario and system were different, many participants mentioned that they could remember what they had done in other spaces experienced earlier in the experiment and made similar decisions for new spaces. Applying more complex methods to distract participants from using their spatial memory could be helpful. Additionally, it may be beneficial to allow future participants more freedom to manipulate the design objects. Changing the wall height and rotating objects should both be explored.

In future research, real-world full-scale spaces should be included in the experiments as a third system. Adding this extra element would produce another branch of data that could be used to test the accuracy of users' spatial perceptions and the differences in decisions made in the DT system, IVRIE, and the real world.

**Author Contributions:** Conceptualization, S.A. and A.R.; methodology, S.A. and A.R; validation, S.A.; formal analysis, S.A.; data curation, S.A.; visualization, S.A.; writing—original draft preparation, S.A.; writing—review and editing, S.A. and A.R. All authors have read and agreed to the published version of the manuscript.

**Funding:** This research received no external funding.

**Institutional Review Board Statement:** The study was conducted in accordance with the Declaration of Helsinki, and approved by the Institutional Review Board of NORTH CAROLINA STATE UNIVERSITY (protocol code: 21067, approval date: 18 November 2020).

**Informed Consent Statement:** Informed consent was obtained from all subjects involved in the study.

**Data Availability Statement:** The data presented in this study are available on request from the corresponding author.

**Acknowledgments:** Thanks to the students of the North Carolina State University for their generous participation in this study.

**Conflicts of Interest:** The authors declare no conflict of interest.

## Abbreviations

| | |
|---|---|
| VR | Virtual Reality |
| TVR | Traditional Virtual Reality |
| IVRIE | Immersive Virtual Reality Interactive Environment |
| DT | Desktop system |
| 3D | Three-dimensional |

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
