# Peer review of "Understanding the Effects of Virtual Reality System Usage on Spatial Perception: The Potential Impacts of Immersive Virtual Reality on Spatial Design Decisions"

_sustainability, doi:10.3390/su141610326_

Round 1

Reviewer 1 Report

Understanding the Effects of Virtual Reality System Usage on Spatial Perception

The considered manuscript is dedicated to comparison of Virtual Reality Interactive Environment (IVRIE) and traditional Virtual Reality systems with respect to human spatial perception / cognition of spatial factors. The authors particularly focus on spatial design decisions made by users of such systems.
I find the research problem quite novel and practical. The authors describe the theoretical base in good detail and perform experimental study with representative users. The paper itself is well structured and the language is good. I recommend accepting it for Sustainability, but suggest that the authors make some minor improvements - see below.

Most importantly, I would recommend introducing the Discussion section. There, the authors should:
1) List limitations of their research, which are currently missing, including methodological ones. For instance, how confident are the authors in attributing the differences in design decisions to spatial perception?
2) Provide a more conceptual outlook of the results with respect to human spatial perception and biases. I do understand that the study focuses on design, but higher-level take-aways would be nice.
3) Provide practical guidelines for building such design systems. For instance, which biases or advantages are involved in IVRIE vs. DT? In which designers could come closer to an "ideal" or a baseline design solution (possibly obtained in a traditional, non-VR system)?
4) Discuss the results, comparing them with the state-of-the-art (also see below).

The second problem I find with the manuscript is that the references need to be updated and more recent sources included in the review and the discussion.
Currently, out of 41 references only 9 references are within 5 last years, and only 2 references are within 3 last years.

Some misc recommendations:
1) Since the authors make multiple comparisons, they might consider using a statistical correction.
2) The authors should consider including the means when they report statistical differences.
3) 736: "ThThe (*) symbol" - mistype. Also, several lines are blank between the table and this text

Reviewer 2 Report

I congratulate the authors for choosing the field of research (virtual reality) which is a pioneering one and offers quite a lot to explore. Expectations are thus created that are not fully reflected in the research methodology chosen and the method of disseminating the results. To understand the scientific approach, a presentation of the sources of the figures and tables would be welcome! Also, a detail of how this research contributes to the development of knowledge in the field would be welcome!

Reviewer 3 Report

The presented work approaches an interesting and novel topic, the influence of virtual environments on spatial perception and how different VR systems, desktop based vs. Immersive VR using HMD, influence the user's performance.

The paper is well written following a logical, scientific adequate structure presenting a well-explained problem based on a good literature review. Related work references are suitable motivating the research necessity. 

Experiments presented in this work are convincing, event though two different software were used, one for DT-VR and one for IVR, the authors have successfully highlighted the similarities between the two and restricted the participants to use command that were identical in both SW making this a valid experimental procedure. Moreover, the authors supplied sufficient information for replicating the experiment.

Further, there are a few recommendations which might improve the scientific value of this research:

- As stated in chapter 2, presence is one of the key features of virtual environments. During the presented experiment, when users are exploring from an eye-level in IVR, could the participants see a virtual representation of their limbs? does visualising the limbs increase the presence? Also, at line 119 it would be indicated to cite the sources used.

- It would be interesting to see if age and gender have any influence on the presented experiment. Has the sample population been equally balanced?

- The interaction device for the IVRIE is composed by the VR controllers or smart gloves? (Line 307) Please explain.

Format and content observations:

-line 255 introduces an acronym which is not defined (even though it is quite self-explanatory)

-line 302 -> "3600 viewshed"

-line 324 "IVREA" please explain acronym 

- in fig. 1 at stage 1 there is stated "IVRIE features vs TVR" while in fig. 2 TVR is referred as DT. Consistency is recommended.

- Fig. 4 shows two scenarios each consisting in 4 3D models, for each scenario it is not clear which are the differences between corridor 1 and 2 respectively enclosure 1 and 2. Is one used for DT and one for IVR? Please explain. Also, dimensional constraints over the models are recommended. 

Round 2

Reviewer 2 Report

The determination of the authors to carry out an applied research is to be appreciated. Didactically, the article has potential. We encourage authors to develop future implications for the environment and sustainable development.